# Functional 14-3-3 Proteins: Master Regulators in Plant Responses to Salt Stress

**DOI:** 10.3390/plants14233568

**Published:** 2025-11-22

**Authors:** Dongxue Tang, Yiwu Zhao, Zhongliang Wang, Junwen Kong, Naiqing Dong, Ling Zheng, Shuangshuang Zhao

**Affiliations:** 1Shandong Provincial Key Laboratory of Plant Stress Biology and Genetic Improvement, Life Science College, Shandong Normal University, Jinan 250014, China; 2Institute of Chinese Medicine Resources, Shandong Academy of Chinese Medicine, Jinan 250014, China

**Keywords:** 14-3-3 proteins, salt stress, protein kinases, ion channels, transcription factors

## Abstract

14-3-3 proteins are highly conserved regulatory molecules that play a central role in plant responses to salt stress. These proteins modulate the activity, stability, and localization of diverse target proteins. This review summarizes current advances in understanding the multifaceted roles of 14-3-3 proteins in salt stress signaling. Specifically, it details how 14-3-3 proteins interact with and regulate diverse components, including protein kinases, phosphatases, ion channels and transporters, proton pumps, metabolic enzymes, and transcription factors. These interactions are predominantly phosphorylation-dependent and often involve calcium (Ca^2+^) and other second messengers. Additionally, 14-3-3 proteins themselves are subject to post-translational regulation, such as phosphorylation and ubiquitination, which fine-tune their stability and activity under stress conditions. This review highlights 14-3-3 proteins as versatile molecular switches in salt stress signaling, integrating diverse signals to orchestrate stress tolerance mechanisms. It also identifies critical knowledge gaps and outlines future research directions aimed at leveraging these proteins for improving crop resilience to salinity stress, an ongoing challenge in modern agriculture.

## 1. Introduction

The 14-3-3 protein family was first identified as a group of abundant acidic proteins in bovine brain tissue, with the name derived from its fractionation on DEAE-cellulose chromatography and migration pattern during starch-gel electrophoresis. This highly conserved protein family is found across all eukaryotes and plays a critical role in a range of cellular processes. In plants, the 14-3-3 protein family exhibits remarkable diversity and is represented by multiple isoforms, including 13 members in *Arabidopsis thaliana* [1], 8 in rice [2], 25 in cotton [3], 18 in soybean [4], 10 in rubber [5], 14 in *Populus trichocarpa* [6], 8 in foxtail millet [7], 9 in common bean [8], 12 in tomato [9], 21 in *Brassica rapa* [10], 7 in *Brachypodium distachyon* [11], 25 in banana [12], 28 in maize [13], 6 in sorghum [13], 17 in tobacco, 5 in barley and 9 in *Nitraria sibirica* [14]. These isoforms form homo- and heterodimers and are generally classified into two major groups: the ε group and the non-ε group, based on their amino acid sequence similarity [15,16]. Despite their high sequence conservation, mounting evidence suggests that individual 14-3-3 isoforms exert distinct regulatory roles in an isoform-specific manner.

Functionally, 14-3-3 proteins interact with their binding partners through three well-defined phosphorylation-dependent binding motifs, including RSXpSXP [17], RX(Y/F)XpSXP, and (pS/pT)X1-2-COOH [18]. These interactions modulate the activity, localization, stability, or interactions of target proteins. As versatile molecular adaptors, 14-3-3 proteins interact with various cellular components, including kinases, such as WEE1 [19], MPK6 [20], WPK4 [21], phosphotases, transcription factors, and enzymes. Their functional repertoire includes scaffolding for the assemble of large protein complexes, altering protein activity, protecting target proteins from proteolysis, regulating ligands stability and substrate subcellular localization. These broad regulatory roles underline the importance of 14-3-3 proteins across cellular pathways.

Salinity is a major environmental stress that affects plant growth and development. Salt stress occurs due to the accumulation of sodium ions (Na^+^) and chloride ions (Cl^−^) in plant tissues, which leads to intracellular osmotic imbalance and ion toxicity. Plant responses to salinity can be broadly categorized into early-stage osmotic stress, late-stage ion toxicity, and secondary stress, including oxidative damage [22,23]. Early salt stress is caused by the high concentration of salt ions in the periphery of the root, which inhibits root water absorption, root and leaf growth, cell elongation, and the development of new tissues. Long-term salt stress leads to excessive accumulation of Na^+^ and Cl^−^ in plant cells. This affects diverse aspects of cellular activities, such as nutritional imbalance, oxidative stress and reactive oxygen species (ROS) accumulation, damage to membrane structure, metabolic disorder, delay of cell division and growth [23,24]. Both Na^+^ and Cl^−^ can be toxic when accumulated at high concentrations, disrupting enzyme activities, impairing photosynthesis through chlorophyll degradation and stomatal closure, and inhibiting various metabolic processes [22,23]. The cellular disruptions impair critical physiological functions such as photosynthesis, protein synthesis, energy metabolism, and cell division, ultimately leading to stunted plant growth and reduced survival under salt stress [22,23,25,26,27].

To combat salt stress, plants deploy molecular signaling networks that include various secondary messengers such as Ca^2+^, ROS, inositol phosphate, and phytohormones [27,28,29]. Ca^2+^ signaling is particularly crucial, as it activates the Salt Overly Sensitive (SOS) pathway, a key mechanism regulating sodium ion homeostasis. Such signal cascades include Ca^2+^ binding protein perception, protein phosphorylation and dephosphorylation, phospholipid metabolism, etc. [30].

The SOS pathway includes SOS3/SCaBP8 (Ca^2+^ sensors), SOS2 (a protein kinase), and SOS1 (a plasma membrane Na^+^/H^+^ antiporter). Under salt stress conditions, SOS3 and SCaBP8 function as the Ca^2+^ receptors to perceive the salt-induced increases of [Ca^2+^]cyt and then to activate SOS2 kinase activity, which further phosphorylates and activates SOS1 [27,31,32,33,34]. The activation of SOS pathway is specific to NaCl stress and not triggered by other osmotic stressors such as KCl or mannitol in *Arabidopsis thaliana* [34]. Structural analysis of SOS2 has revealed that its activation is regulated by the disassociation of its FISL motif, which interacts with SOS3 under normal conditions. However, the full regulatory mechanism governing SOS2 activity remains unclear [35].

Emerging evidence highlights the critical involvement of 14-3-3 proteins in mediating plant responses to salt stress. As regulatory proteins involved in phosphorylation-dependent signaling, 14-3-3 proteins interact with kinase cascades, ion channels, and transcriptional machinery to transduce stress signals [15]. These interactions enable plants to fine-tune stress responses, including regulation of the SOS signaling pathway. Moreover, 14-3-3 proteins contribute to broader abiotic stress adaptation mechanisms by modulating processes such as ion transport, ROS scavenging, and gene expression regulation.

In this review, we explore the multifaceted roles of the 14-3-3 protein family in plant salt stress responses. We discuss the molecular mechanisms underlying their interactions with phosphorylated targets and their potential contributions to enhancing salt tolerance. By synthesizing current knowledge, we aim to provide insights into the regulatory networks governed by 14-3-3 proteins and their significance in improving plant resilience to salinity.

## 2. 14-3-3 Proteins Regulate Salt Stress-Related Proteins

The 14-3-3 family of regulatory proteins plays diverse roles in plant responses to abiotic stress, including salt stress. These proteins interact with and regulate the activity of multiple targets, such as protein kinases, phosphatases, ion channels, proton pumps, enzymes, and transcription factors (Table 1). Below, we systematically review the role of 14-3-3 proteins in modulating the activity of salt stress-related proteins across several functional categories.

**Table 1 plants-14-03568-t001:** Proteins interacting with 14-3-3 proteins in salt stress and their regulatory mechanisms.

Proteins	Key Regulatory Features	Species	Stress	References
SOS2	Context-dependent: Binding inhibits kinase activity normally. Salt stress induces 14-3-3 degradation, releasing SOS2 for activation.	*Arabidopsis thaliana*	Salt stress	[36,37]
PKS5	Direct Inhibition: Ca^2+^-activated 14-3-3 binds and represses PKS5 kinase activity, releasing its inhibition of SOS2.	*Arabidopsis thaliana*	Salt and alkaline stresses	[37,38]
CPK21	Positive Feedback: Salt-induced Ca^2+^ and autophosphorylation enhance 14-3-3 binding, which further amplifies kinase activity.	*Arabidopsis thaliana*	Salt stress	[39]
TMKP1	Direct Activation: 14-3-3 binding to phosphorylated C-terminus stimulates phosphatase activity, boosting antioxidant defense.	*Triticum durum*	Salt stress	[40]
GORK	Indirect Activation: Activated by CPK21 (which is enhanced by 14-3-3). 14-3-3 may stabilize the CPK21-GORK complex.	*Arabidopsis thaliana*	Salt stress	[39]
TPK1/KCO1	Phospho-Dependent: Phosphorylation (e.g., by CPK3) enhances 14-3-3 binding, increasing channel activity for vacuolar K^+^ release.	*Arabidopsis thaliana*	Salt stress	[41,42]
GLR3.7	Phospho-Switched: Normally inhibited. Salt-induced phosphorylation enhances 14-3-3 binding, modulating its Ca^2+^ channel activity.	*Arabidopsis thaliana*	Salt stress	[43]
SOS1	Direct inhibition: via binding to SOS1 C-terminal.	*Arabidopsis thaliana*	Salt stress	[44]
H^+^-ATPase	Phospho-Binding: 14-3-3 binds phosphorylated C-terminal autoinhibitory domain, displacing it to activate H^+^ pumping.	*Arabidopsis thaliana*	Salt and alkaline stresses	[38,45,46,47]
N/AINV	Enzyme Activation: Interaction enhances sucrose hydrolase activity, increasing osmolyte (glucose/fructose) production.	*Gossypium hirsutum*	Salt and drought stresses	[48]
PLC1	Dual Role: 14-3-3 binding both activates enzyme activity and stabilizes the protein by inhibiting its ubiquitination.	*Oryza sativa*	Salt stress	[49]
VP1	Pump Activation: Interaction enhances H^+^-pyrophosphatase activity, strengthening the proton gradient for vacuolar Na^+^ sequestration.	*Nitraria sibirica*	Salt stress	[14]
bZIP23/62/71	TF Co-activator: Binding enhances transcription factor stability/DNA-binding, upregulating ABA-responsive genes.	*Oryza sativa*/ *Brachypodium distachyon*	Salt, drought, osmotic stresses	[50,51,52]
AREB	ABA Signaling: Interacts with AREB transcription factors to strengthen ABA signaling and stress-responsive gene expression.	*Malus domestica*	Salt, drought stresses	[53]
VSF-1	Nucleo-Cytoplasmic Shuttling: Binding retains TF in cytoplasm. Dephosphorylation under stress releases it for nuclear translocation.	*Solanum lycopersicum*	Hypo-osmotic stress	[54,55,56,57,58]
MYB64	TF Co-activator: Interaction enhances the transcriptional activation function of MYB64.	*Triticum aestivum*	Salt stress	[59]
WRKY18	Stability and Activity: Phospho-dependent interaction enhances TF stability and transcriptional activity of SOS pathway genes.	*Malus domestica*	Salt stress	[60]
ASMT1	Enzyme Recruitment: Phosphorylated 14-3-3 shows enhanced association with ASMT1, promoting melatonin biosynthesis.	*Malus domestica*	Salt stress	[61]
GCN4	Proteasomal Degradation: GCN4 promotes the degradation of specific 14-3-3 isoforms, inhibiting H^+^-ATPase and closing stomata.	*Arabidopsis thaliana*	Drought stress	[62]
RNF1/2	Ubiquitination Link: 14-3-3 protein interacts with E3 ubiquitin ligases RNF1 and RNF2, suggesting potential regulation of 14-3-3 stability.	*Setaria italica*	Salt stress	[63]

### 2.1. Reprogramming Stress Responses by Regulating Signaling Components

#### 2.1.1. Regulation of Protein Kinases

The discovery of the first cDNA sequences for higher plant protein kinase in 1989 marked a major milestone in the study of plant signaling transduction [64]. Protein-serine/threonine kinases constitute an extensive network in plant cells, functioning as a “Central Processor Unit” (CPU) that integrates input signals from environmental stimuli, phytohormones, and other external factors to generate adaptive responses, such as changes in metabolism, gene expression, and cell growth and division [65]. In plants, there are reports on the presence of CBL-interacting protein kinases (CIPKs) and Ca^2+^-dependent protein kinases (CDPKs) (Figure 1).

14-3-3 proteins regulate CIPKs such as SOS2 and SOS2-Like Protein Kinase 5 (PKS5) in response to salt stress. The SOS pathway regulates the intracellular sodium ion homeostasis and salt tolerance in plants. Upon exposure to salt stress, the Ca^2+^-binding protein ANN4 mediates salt-induced Ca^2+^ signaling, triggering a rapid rise in cytosolic [Ca^2+^] that is essential for activating the SOS pathway [66]. SOS2, a CIPK, functions as the central regulator in this pathway, and its kinase activity is fine-tuned under both normal and salt stress conditions. Under normal conditions, 14-3-3 proteins, specifically isoforms λ and κ, interact with SOS2 to suppress its kinase activity, ensuring the SOS pathway remains minimally active in the absence of salt stress [36]. However, during salt stress, the interaction between 14-3-3 proteins and SOS2 is disrupted, leading to activation of the SOS pathway for salt tolerance. Further studies demonstrate that salt stress promotes ubiquitination and 26S proteasome-dependent degradation of 14-3-3λ and κ proteins [67]. Interestingly, this salt-induced degradation of 14-3-3 proteins was detected in wild-type plants but not in *scabp8* mutant. This indicates that SCaBP8 plays a regulatory role in maintaining 14-3-3 protein stability under salt stress. While SCaBP8 and SOS3 are both Ca^2+^-binding proteins associated with the SOS pathway, SCaBP8 appears to play a more prominent role in regulating 14-3-3 protein stability under salt stress. The mechanism underlying this distinction remains elusive. Investigating how SCaBP8 mediates 14-3-3 degradation during salt stress will provide valuable insights into the differential roles of plant Ca^2+^-binding proteins in stress adaptation.

In common bean (*Phaseolus vulgaris*), the 14-3-3 isoforms GF14a and GF14g are linked to salt tolerance, possibly by interacting with SOS2, a key component of the SOS signaling pathway [68]. The SOS pathway maintains sodium ion homeostasis and salt tolerance in plants by modulating the activity of the Na^+^/H^+^ antiporter (SOS1), which is activated by the kinase SOS2 during stress conditions. Recent studies indicate that Ca^2+^-activated 14-3-3 proteins function as molecular switches in salt stress tolerance, mediating dynamic regulation of SOS and related kinases such as PKS5. Under normal growth conditions, PKS5 can interact with and phosphorylate SOS2, promote the interaction between SOS2 and 14-3-3 proteins, and repress SOS2 activity to limit SOS1 Na^+^/H^+^ antiporter activity to basal levels. Upon salt stress, 14-3-3 proteins decode a salt-induced Ca^2+^ signal. This promotes an interaction between 14-3-3 proteins and PKS5, which represses PKS5’s kinase activity. Consequently, the inhibition of SOS2 is released, allowing for the activation of PM H^+^-ATPase and SOS1. These findings indicate that a salt-induced Ca^2+^ signal is first decoded by the 14-3-3 and SOS3/SCaBP8 proteins. These proteins then selectively activate or inhibit the downstream kinases SOS2 and PKS5. Ultimately, this coordinated regulation modulates the activities of the plasma membrane Na^+^/H^+^ antiporter and H^+^-ATPase to maintain Na^+^ homeostasis [37]. However, it is not clear whether and how Ca^2+^ and Ca^2+^-binding protein SCaBPs/CBLs play roles in regulating PKS5 and SOS2 under normal growth conditions, and whether and how they function in repression of PKS5 under salt stress.

The compartmentalization of Na^+^ within plant cells is another means to mitigate salt stress, where plants sequester excess Na^+^ into organelles such as vacuoles to rapidly reduce Na^+^ concentration in cytoplasm [69,70]. This process is facilitated by vacuole membrane Na^+^/H^+^ transporter (Na^+^/H^+^ exchanger) which transfers redundant Na^+^ into the vacuoles. This mechanism is driven by the proton gradient mediated by membrane H^+^-ATPase and H^+^-pyrophosphatase [71,72]. Interestingly, 14-3-3 proteins may contribute to Na^+^ compartmentalization by mediating H^+^-ATPase activity upon salt or alkaline stress. Additionally, 14-3-3 proteins regulate PKS5 in response to alkaline stress, where PKS5 and the chaperone DNAJ HOMOLOG3 (J3) play important roles in facilitating H^+^ efflux by regulating the interaction between plasma membrane (PM) H^+^-ATPase and 14-3-3 proteins in *Arabidosis thaliana* [38,73]. In tomato, the expression levels of 14-3-3 isoforms such as *TFT1*, *TFT4*, *TFT6*, and *TFT7* are elevated under alkaline stress, with *TFT4*-overexpression in *Arabidopsis* significantly enhancing plant growth under such conditions. Further analysis indicates that PM H^+^-ATPase-mediated H^+^ secretion, basipetal auxin transport, and the PKS5-J3 pathway collectively contribute to the TFT4-mediated response to alkaline stress [74]. Previous studies have also shown that PKS5 interacts with the Ca^2+^-binding protein SCaBP1, and that high external pH conditions can trigger an increase in cytosolic free Ca^2+^ concentration. This suggests that PKS5 may be part of a Ca^2+^-signaling pathway involved in PM H^+^-ATPase regulation [38]. However, whether 14-3-3 proteins directly interact with Ca^2+^-binding proteins and play a role in Ca^2+^-signaling pathway under alkaline stress conditions remains to be elucidated. Future investigations are necessary to clarify the mechanisms underlying the role of 14-3-3 proteins in this process.

14-3-3 proteins regulate CDPKs, such as CPK21, in response to salt stress. Salt stress triggers plasma membrane depolarization and a subsequent increase in cytosolic Ca^2+^ concentrations, which in turn, activate the CPK21, enabling its autophosphorylation [75,76,77]. This autophosphorylation enhances CPK21 capacity to interact with 14-3-3 proteins, which further amplifies its kinase activity. Once activated, CPK21 phosphorylates specific residues (T344, S518, and S649) on the GORK K^+^-channel, leading to channel opening and facilitating K^+^ efflux. Additionally, 14-3-3 proteins may maintain GORK channel activity by stabilizing the CPK21-GORK protein complex during this process [39]. Thus, 14-3-3 proteins fine-tune kinase activity to modulate ion homeostasis under salt stress.

#### 2.1.2. Regulation of Protein Phosphatases

Mitogen-activated protein kinase phosphatases (MKPs) are important negative regulators in the MAPK signaling pathways, which play crucial roles in controlling plant growth, development, and responses to stresses [78,79,80]. In durum wheat (*Triticum turgidum* susp. *durum*), a specific MKP, TMKP1, acts as a positive regulator of salt stress tolerance, primarily through its role in enhancing antioxidant enzyme activities [81]. TMKP1 contains a 14-3-3-binding motif and interacts with 14-3-3 proteins in vivo, which stimulates its phosphatase activity in a phosphorylation-dependent manner [40]. Further studies are needed to elucidate the physiological significance of the TMKP1/14-3-3 interaction in MAPK signaling during stress responses.

### 2.2. Maintaining Cellular Balance by Regulating Ion Homeostasis and Membrane Transport

#### 2.2.1. Regulation of Ion Channels

14-3-3 proteins regulate various ion channels, including potassium channels such as TPK1 and GORK, the nonselective cation channel GLR3.7, and the Na^+^/H^+^ antiporter SOS1 in response to salt stress.

Potassium (K^+^) release from vacuoles under salt stress is essential for maintaining cytosolic K^+^ homeostasis, as well as preserving cellular turgor and membrane potential [82,83,84]. Potassium channel TPK1 is regulated by 14-3-3 binding, which is enhanced by phosphorylation of Ser42 in its N-terminus. Salt stress elevates cytosolic Ca^2+^, activating TPK1 and promoting K^+^ release from vacuoles. The stress-activated kinase CPK3 phosphorylates TPK1 Ser42, stabilizing 14-3-3 interactions and sustaining K^+^ efflux under prolonged stress [41,42]. This dual regulation ensures K^+^ homeostasis under fluctuating stress conditions.

The mechanism underlying the elevation of cytoplasmic Ca^2+^ levels following salt stress involves multiple pathways, including glutamate receptor-like channels (GLRs). Previous studies have identified GLR2 as contributors to Ca^2+^ influx in plant cells [85]. The GLR3.7 is inhibited by 14-3-3ω under normal conditions. Salt stress induces phosphorylation of GLR3.7 at Ser860 by CDPKs (CDPK3, CDPK16, and CDPK34), enhancing its interaction with 14-3-3ω and modulating Ca^2+^ channel activity, which influences downstream stress responses [43]. However, specific changes in GLR3.7 ion permeability under salt stress need further verification.

SOS1 is the core plasma membrane Na^+^/H^+^ antiporter in plant salt stress response. It maintains ion homeostasis by exporting excess intracellular Na^+^ out of cells, and its activity is regulated by the SOS signaling pathway, wherein phosphorylation of the C-terminal autoinhibitory region by SOS2 relieves inhibitory and activates its transport function [86]. SOS1 interacts with 14-3-3 proteins via a non-canonical binding site in its C-terminal region. Overexpression of the SOS1 C-terminus sequesters 14-3-3 proteins, activating Na^+^ efflux, reducing Na^+^ accumulation, and enhancing salt tolerance [44]. The precise effect of 14-3-3 binding on SOS1 transport activity and the functional differentiation among 14-3-3 isoforms require further investigation.

#### 2.2.2. Regulation of Proton Pumps

The regulation of proton pumps, notably the plasma membrane H^+^-ATPase and the vacuolar H^+^-pyrophosphatase, by 14-3-3 proteins is a pivotal mechanism in plant adaptation to salt stress.

H^+^-ATPase, a key enzyme localized in the plasma membrane, is essential for providing the energy required for ion and nutrient transport by generating and maintaining proton gradients. The activity of H^+^-ATPase is regulated by its interaction with 14-3-3 proteins. Specifically, 14-3-3 proteins bind directly to the C-terminal autoinhibitory domain of H^+^-ATPase, leading to displacement of the autoinhibitory region and subsequent activation of the enzyme [45,87]. This activation is typically triggered by the phosphorylation of a specific residue, Thr948, within the C-terminal region of H^+^-ATPase, which creates a high-affinity binding site for 14-3-3 proteins [88].

In plants, PM H^+^-ATPase are also one of the most critical regulators in responding to hormones and environmental stimuli, including ABA, Ca^2+^ signaling, and increased soil salinization. As the master enzyme of cellular activities in plant cells, the activation of PM H^+^-ATPase is strictly regulated in plant growth and development. Changes in the phosphorylation levels of many amino acid residues result in changes in the interaction between 14-3-3 proteins and the C terminal region of the PM H^+^-ATPase, thereby activating or inhibiting its enzymatic activity. For example, phosphorylation of *Arabidopsis* PM H^+^-ATPase AHA1 residue Thr948 and AHA2 residue Thr947 enhanced the interaction between 14-3-3 proteins and its C-terminal domain, displacing the autoinhibitory region and thereby enhancing its enzyme activity [46]. Conversely, the SOS2-LIKE PROTEIN KINASE5 (PKS5) negatively regulates salt–alkaline tolerance by phosphorylating Ser931 in AHA2, which disrupts the interaction between 14-3-3 proteins and AHA2. This dissociation inhibits PM H^+^-ATPase activity, thereby reducing proton pumping efficiency and affecting the plant’s response to salt stress in *Arabidopsis thaliana* [38]. Additionally, lysine-ε-acetylation of 14-3-3λ/GRF6 at Lys56 inhibits its binding to AHA2, reducing H^+^-ATPase activity. Under alkaline stress, reduced GRF6 acetylation enhances H^+^-ATPase activity and plant adaptability [47,89]. The regulatory enzymes and mechanisms controlling GRF6 acetylation require further study. Whether similar phosphorylation and acetylation mechanisms regulate H^+^-ATPase in other plant species remains unknown.

In the salt-tolerant shrub *Nitraria sibirica*, Ns14-3-3 5a interacts with vacuolar H^+^-pyrophosphatase (NsVP1), enhancing its activity and H^+^-pumping to drive Na^+^ compartmentalization. Tissue-specific expression (upregulated in shoots, downregulated in roots) redirects Na^+^ storage, reducing root toxicity [14,90,91,92,93]. The exact interaction site and regulatory mechanism of NsVP1 require further study.

### 2.3. Coordinating Metabolism and Gene Expression by Regulating Metabolic Enzymes and Transcription Factors

#### 2.3.1. Regulation of Key Metabolic Enzymes

14-3-3 proteins regulate the activity of several key enzymes involved in plant responses to salt stress. These include sucrose hydrolase N/AINV13, phospholipase C1 PLC1, and N-acetylserotonin methyltransferase ASMT1 (ASMT1 is further discussed in Section 3.1).

Neutral/alkaline invertases (N/AINVs), a key subgroup of sucrose hydrolases, play vital roles in hydrolyzing sucrose into glucose and fructose, thereby regulating plant carbon allocation, cell differentiation, and stress responses. N/AINVs are known to be involved in plant responses to drought and salt stress in species such as *Arabidopsis thaliana* and *Oryza sativa* [94,95,96,97,98]. In cotton, GhN/AINV13 interacts with 14-3-3 isoforms (GhGF12, GhGF13, GhGF14, GhGF22), enhancing its hydrolase activity. This promotes glucose and fructose production, maintaining osmotic pressure and reducing oxidative damage [48].

Phospholipase C (PLC), a central enzyme in plant phospholipid signaling, comprises phosphatidylinositol-specific PLC (PI-PLC) and non-specific PLC (NPC). PI-PLC hydrolyzes phosphatidylinositols (e.g., PI, PI(4)P, and PI(4,5)P_2_) into diacylglycerol (DAG) and inositol phosphates (IP_3_), which—along with their derivatives such as phosphatidic acid (PA) and IP_6_—act as key second messengers regulating plant growth, development, and stress responses [99,100,101]. In rice, the isoforms phospholipase C1 (OsPLC1) and phospholipase C4 (OsPLC4) are particularly crucial for salt stress tolerance. These enzymes exert their effects by inducing stress-responsive Ca^2+^ signals or phosphatidic acid (PA) signals [102,103]. OsGF14b interacts with OsPLC1, activating its hydrolytic activity and inhibiting its ubiquitination under salt stress. This enhances production of lipid second messengers (e.g., IP_3_, PA), promoting downstream stress responses [49]. The E3 ubiquitin ligase for OsPLC1 and the mechanism by which OsGF14b blocks ubiquitination require further study.

#### 2.3.2. Regulation of Transcription Factors

Studies have shown that 14-3-3 proteins interact with various stress-responsive transcription factors, including members of the bZIP, MYB, and WRKY families. These interactions modulate the transcriptional activity of them, thereby enabling the plants to adapt to salt stress.

Both salt and drought stresses induce osmotic stress, changes in intracellular ROS, and fluctuations in ion concentration. These stresses share significant overlaps in the transcription factors involved in their respective stress responses. 14-3-3 proteins have been shown to interact with ABRE-binding transcription factors (ABFs/AREBs), which are key components of ABA signaling pathway. ABFs belong to the bZIP transcription factor family and regulate a variety of stress-responsive processes. In *Brachypodium distachyon*, BdGF14d and BdGF14a interact with bZIP transcription factors (e.g., BdbZIP62, BdbZIP71), strengthening ABA signaling, ROS scavenging, and ion transport under salt stress [50,51]. Similarly, in apple, MdGRF11 interacts with MdAREB transcription factors, enhancing stress tolerance through ABA signaling [53]. In rice, OsGF14f interacts with OsbZIP23, enhancing its DNA-binding and transcriptional activity, upregulating stress-responsive genes (e.g., *OsLEA3-2*), and forming a positive feedback loop to improve osmotic stress tolerance [104]. Conversely, OsGF14b acts as a negative regulator [52]. Another Brachypodium gene, *BdGF14g*, responds to ABA, H_2_O_2_, and PEG treatment. It encodes a protein that interacts with NtABF2, potentially increasing endogenous ABA levels and upregulating ABA-related genes such as *NtNCED1* and *NtERD10C* under drought. This suggests that BdGF14g enhances NtABF2’s DNA-binding to reinforce ABA-dependent transcription—though further validation is needed [105].

14-3-3 binding often depends on transcription factor phosphorylation and can influence subcellular localization. For example, tomato bZIP transcription factor VSF-1 and *Arabidopsis* VIP1 dissociate from 14-3-3 proteins under hypo-osmotic stress, translocating to the nucleus, while hyper-osmotic stress inhibits this translocation [54,55,56,57,58]. Overexpression of VSF-1 enhances sensitivity to non-ionic osmotic stress, suggesting its function as a negative regulator [58].

Interactions with MYB and WRKY transcription factors also contribute to salt tolerance. In wheat, TaGRF6-A interacts with TaMYB64, enhancing its transcriptional activity and stress gene expression [59]. In apple, MdGRF8 interacts with phosphorylated MdWRKY18 (Ser179), stabilizing it and boosting its transcriptional activity, which upregulates *MdSOS2* and *MdSOS3* expression [60]. These interactions highlight the role of 14-3-3 proteins in fine-tuning transcriptional networks under salt stress.

## 3. Regulation of 14-3-3 Proteins in Response to Salt Stress

14-3-3 proteins play an extremely important role in plant responses to abiotic stresses. They function as upstream regulators, modulating the activity, stability, and cellular localization of diverse target proteins critical to stress adaptation. Interestingly, the function and stability of 14-3-3 proteins themselves are tightly regulated by upstream proteins through various post-translational modifications, such as phosphorylation and ubiquitination, which fine-tune their roles during stress responses (Table 1). This section systematically summarizes current understanding of how phosphorylation and ubiquitination modulate 14-3-3 proteins during salt stress.

### 3.1. Regulation of 14-3-3 Proteins by Phosphorylation

Phosphorylation represents a major regulatory mechanism for 14-3-3 proteins during salt stress, mediated notably by CDPKs and receptor-like cytoplasmic kinases (RLCKs). CDPKs are unique Ca^2+^ sensors integrating kinase and calmodulin-like domains within a single polypeptide. Beyond phosphorylating 14-3-3 client proteins, certain CDPKs directly modify 14-3-3 proteins, although direct evidence in the context of plant salt stress remains limited.

In rice, OsCPK21 was shown to interact with OsGF14e (an Os14-3-3 isoform) and phosphorylate it at Tyr138. Overexpression of *OsCPK21* or wild-type *OsGF14e* enhanced the expression of ABA- and salt-responsive genes, whereas mutation of the phosphorylation site (OsGF14e Y138A) attenuated this response, suggesting that OsCPK21 may relay signals to transcription factors via post-translational modification of OsGF14e during ABA and salt stress [106]. Separately, OsCDPK1 was found to upregulate *GF14c* expression. Overexpression of either *OsCDPK1* or *GF14c* improved drought tolerance in rice, with OsCDPK1 acting as a sugar starvation- and GA-inducible kinase that suppresses GA biosynthesis while activating GF14c to promote stress adaptation [107]. Further identification of OsCDPK1 substrates that interact with GF14c will help elucidate the mechanistic basis of this drought tolerance pathway.

RLCKs transduce transmembrane signals in response to diverse stresses, including salinity [108]. In apple, MdPBL34, an RLCK, enhances salt tolerance. Knockdown of *MdPBL34* led to chlorophyll degradation, elevated ROS and MDA accumulation, and increased salt sensitivity. Salt stress rapidly activated MdPBL34, which then interacted with and phosphorylated the 14-3-3 protein MdGRF10 at six C-terminal residues (Ser199, Thr214, Ser219, Tyr220, Ser223, Thr224) [61].

Notably, phosphorylation of MdGRF10 by MdPBL34 promotes its association with MdASMT1, a key enzyme in melatonin biosynthesis. Under salt stress, MdASMT1 induction elevates melatonin levels, which scavenges ROS, mitigates oxidative membrane damage (reflected by reduced MDA), maintains photosynthetic efficiency, and thereby improves salt tolerance [109,110,111]. Transgenic apple materials overexpressing *MdASMT1* confirmed these effects, underscoring a phosphorylation-regulated 14-3-3 module in salt adaptation [61].

### 3.2. The Regulation of 14-3-3s by Ubiquitination

Ubiquitination also plays a critical role in regulating 14-3-3 protein stability and function during stress. In *Arabidopsis*, GENERAL CONTROL NONREPRESSIBLE4 (AtGCN4), an AAA^+^-ATPase, promotes drought tolerance by facilitating proteasome-mediated degradation of RIN4 and specific 14-3-3 proteins. This degradation attenuates plasma membrane H^+^-ATPase activity, leading to stomatal closure [62]. *AtGCN4*-overexpressing plants also exhibited restricted pathogen entry, suggesting that 14-3-3-mediated stomatal regulation contributes to both abiotic and biotic stress resilience. The identity of the ubiquitin ligase(s) responsible for 14-3-3 ubiquitination in this pathway, however, remains unknown.

In foxtail millet (*Setaria italica*), salt stress significantly upregulates *SiGRF1*. This 14-3-3 protein interacts with the E3 ubiquitin ligases SiRNF1 and SiRNF2, implying potential regulation via the ubiquitin–proteasome system [63]. The precise mechanism and functional consequences of these interactions under salt stress await further investigation.

### 3.3. Concluding Remarks on 14-3-3 Regulation

In summary, 14-3-3 proteins are dynamically regulated by phosphorylation and ubiquitination under salt stress. Phosphorylation, mediated by kinases such as CDPKs and RLCKs, modulates 14-3-3 interaction capabilities and downstream signaling, influencing stress-responsive gene expression and physiological outcomes such as ROS scavenging and osmotic adjustment. Ubiquitination, conversely, primarily affects 14-3-3 protein stability, with implications for stomatal movement and stress signaling. Despite these advances, key questions remain—particularly regarding the identity of E3 ligases involved in 14-3-3 ubiquitination and the full spectrum of kinases that target 14-3-3 proteins under stress. Future studies should aim to delineate these mechanisms across diverse plant species to facilitate the development of crops with enhanced salt tolerance.

## 4. Discussion

The extensive research summarized in this review underscores the pivotal role of 14-3-3 proteins as central regulators in plant salt stress responses. These highly conserved proteins function as molecular scaffolds, adaptors, and modulators, interacting with a diverse array of client proteins, including kinases, phosphatases, ion transporters, metabolic enzymes, and transcription factors [15,16]. A comprehensive evolutionary perspective reveals that the 14-3-3 gene family has undergone significant expansion, duplication, and contraction events across the plant kingdom, contributing to its functional diversification in stress responses. Recent phylogenetic analyses of 46 angiosperm species, including basal angiosperms like *Amborella*, show that the 14-3-3 family split early into ε and non-ε types, with at least four ancient subfamilies (iota, epsilon, kappa, psi) forming via duplication events in a common ancestor of angiosperms. Subsequent lineage-specific whole-genome duplications (WGDs) and small-scale duplications (SSDs), followed by gene loss, further shaped isoform diversity. For instance, Poaceae lost mu and kappa isoforms, while Fabaceae lost epsilon isoforms, illustrating how evolutionary dynamics contribute to functional specialization and redundancy [112]. These evolutionary dynamics have given rise to distinct subfamilies (ε and non-ε) with specialized roles, such as the non-ε group’s stronger binding affinity for H^+^-ATPases, which is crucial for ion homeostasis under stress [113,114].

A key emerging theme is that 14-3-3 proteins function as molecular switches that repress key signaling kinases under normal conditions and undergo signal-induced reconfiguration to activate stress tolerance mechanisms under salt stress. This “release-of-inhibition” model operates through distinct, yet complementary, mechanisms targeting different nodes within the SOS signaling network. Specifically, under normal conditions, 14-3-3 proteins (such as λ and κ) bind to and inhibit SOS2, while PKS5 phosphorylates SOS2 and further promotes its association with 14-3-3 proteins, collectively maintaining the SOS pathway at basal activity. Upon salt stress, a Ca^2+^ signal is sensed by Ca^2+^-binding proteins including SCaBP8 and SOS3, leading to two coordinated events: first, the disruption of the 14-3-3–SOS2 interaction, which releases SOS2 from inhibition; and second, the promotion of 14-3-3–PKS5 interaction, which represses PKS5 kinase activity and thereby relieves PKS5-mediated inhibition of SOS2 [36,37,67].

Notably, although PKS5 and SOS2 are homologous CIPK family members, they have evolved opposing regulatory relationships with 14-3-3 proteins during stress, which is central to the switch-like mechanism. This functional divergence ensures a robust, multi-layered activation of SOS2: one layer through direct release (SOS2 itself), and another through inactivation of its negative regulator (PKS5). Both mechanisms converge on the full activation of SOS2, which in turn phosphorylates and activates SOS1 and PM H^+^-ATPase to restore ion homeostasis. Thus, the opposing interactions of 14-3-3 proteins with SOS2 (disruption) and PKS5 (promotion) are not contradictory but represent a sophisticated, coordinated strategy to amplify SOS2 output and ensure a robust salt stress response.

The regulation of 14-3-3 proteins is highly complex and isoform-specific. For instance, while OsGF14f acts as a positive regulator of osmotic stress tolerance [104], OsGF14b appears to be a negative regulator [52]. This duality in functionality allows for a sophisticated and context-dependent regulatory network, facilitated by isoform-specific interactions and the ability to form heterodimers. Evolutionary studies further highlight that isoform-specific functions may arise from subfunctionalization or neofunctionalization following gene duplication, as seen in the distinct expression patterns and stress responses of paralogous 14-3-3 genes in species like *Arabidopsis pumila* and *Hordeum vulgare* [112]. These isoform-specific roles are further tuned by precise PTMs, primarily phosphorylation at specific binding motifs of target proteins [17,18]. Nonphosphorylation-dependent mechanisms, such as the interaction between 14-3-3 proteins and the SOS1 C-terminal region, add another layer of complexity to their regulatory functions [44]. Additionally, 14-3-3 proteins themselves undergo PTM, including phosphorylation by kinases like OsCPK21 [106] and MdPBL34 [61], and acetylation, which modulates their activity. For instance, acetylation of GRF6 negatively regulates its ability to activate H^+^-ATPase [47].

Beyond phosphorylation and ubiquitination, post-transcriptional modifications such as alternative splicing represent another critical layer of regulation for 14-3-3 proteins under stress conditions. In mammals and invertebrates, alternative splicing generates 14-3-3 isoforms with distinct functional properties. For example, Han et al. identified a novel splicing variant of human 14-3-3 epsilon (14-3-3 epsilon sv) that lacks the N-terminal α-helix required for dimerization [115]. Despite its monomeric status, this variant retained the ability to inhibit UV-induced apoptosis, demonstrating that dimerization is not always necessary for 14-3-3 function [115]. Similarly, in the shrimp *Litopenaeus vannamei*, two splicing variants of 14-3-3 epsilon (14-3-3EL and 14-3-3ES) were identified, which exhibited tissue-specific expression and differential responses to white spot syndrome virus (WSSV) infection [116]. The 14-3-3EL variant, which retains an intron leading to an extended C-terminus, was upregulated in gill and muscle upon infection, while both variants were downregulated in the lymphoid organ, suggesting that splicing-mediated isoform switching may fine-tune immune responses. Although similar splicing variants have not been extensively characterized in plants, the conservation of this mechanism across kingdoms implies that alternative splicing of 14-3-3 genes could contribute to stress adaptation in plants as well, potentially affecting dimerization capacity, subcellular localization, or partner selectivity.

Despite significant progress, multiple questions remain unanswered. First, the precise mechanism by which Ca^2+^-binding proteins like SCaBP8 and SOS3 differentially regulate 14-3-3 protein stability and target protein interactions is not fully elucidated [67]. Second, while ubiquitination of 14-3-3 proteins is a key mechanism for their removal through proteasomal degradation, the specific E3 ubiquitin ligases involved (e.g., for *Arabidopsis* 14-3-3 proteins or OsPLC1) are largely unknown [49,62,67]. Third, the functional significance of the immense diversity of 14-3-3 isoforms across plant species is poorly understood. Evolutionary genomics approaches, such as comparative phylogenetics and synteny analysis, can help trace the origin of specific isoforms and link gene family expansion/contraction to functional diversification [112]. Determining whether different isoforms have tissue-specific, or stress-specific roles will deepen our understanding of their regulatory networks. Finally, current studies primarily focus on binary interactions. Understanding how 14-3-3 proteins integrate multiple, simultaneous signals to coordinate the assembly and activity of large protein complexes in stress responses remains a major challenge.

## 5. Conclusions and Future Perspectives

In conclusion, 14-3-3 proteins are pivotal integrators within the plant salt stress signaling network. As master regulators, they translate upstream signals—particularly Ca^2+^ bursts—into downstream physiological responses by modulating the activity, stability, localization, and protein–protein interactions of a vast repertoire of stress-related proteins. Their regulatory roles span the entire salt tolerance process. This includes initial sensing and signal transduction, for example, through the regulation of CPK21 and GLR channels [39,43]. It also encompasses the maintenance of ion homeostasis, through the modulation of SOS2, SOS1, H^+^-ATPases, and TPK1 [36,37,41,44]. Furthermore, 14-3-3 proteins are involved in long-term adaptive responses, such as by activating transcription factors like bZIPs and WRKYs [50,60,104]. A common regulatory mechanism employed by 14-3-3 proteins involves a release-of-inhibition process, where salt stress alleviates their constitutive repression of positive regulators of salt stress tolerance.

From an evolutionary standpoint, the 14-3-3 gene family has undergone dynamic expansion and functional diversification across plant lineages, driven largely by WGD and segmental duplication events. The emergence of ε and non-ε subfamilies with distinct expression patterns and client specificities underscores the adaptive evolution of this family in response to environmental stresses. Gene duplication and subsequent lineage-specific loss (e.g., loss of epsilon in Fabaceae, mu and iota in Poaceae) have shaped the present-day diversity of 14-3-3 isoforms, with coexpression analyses revealing that evolutionarily related isoforms can form species-specific functional groups under stress [112]. Comparative studies in Brassicaceae, for example, reveal that purifying selection has predominantly shaped the family, yet ε-group members exhibit relatively relaxed constraints, suggesting their potential for functional innovation [113,114].

Notably, post-transcriptional regulation—especially alternative splicing—emerges as an understudied yet potentially widespread mechanism for generating functional diversity among 14-3-3 isoforms. Studies in non-plant systems have shown that splicing variants can alter dimerization capacity, target specificity, and subcellular localization, thereby modulating stress responses without altering the core protein structure [115,116]. In plants, the existence and functional relevance of such splicing variants remain largely unexplored. Future efforts should aim to systematically identify and characterize 14-3-3 splice variants across plant species, particularly under abiotic stresses such as salinity.

Future research should focus on key areas to deepen our understanding of 14-3-3-mediated regulatory mechanisms and their broader implications for plant stress biology. Specifically, identifying the upstream regulators of 14-3-3 proteins, such as the kinases, ubiquitin ligases, and other components responsible for their activation, degradation, and post-translational modifications, remains a critical gap. Integrating evolutionary genomics with functional studies will help elucidate how gene duplication and neofunctionalization have shaped the present-day diversity and functional specialization of 14-3-3 isoforms. In particular, the role of alternative splicing in generating isoform diversity and functional specialization under stress deserves greater attention. To address functional redundancy among isoforms and dissect isoform-specific roles, emerging techniques such as CRISPR-Cas9–based gene editing for generating isoform-specific mutants, along with isoform-specific interactomics and structural biology approaches, will be essential. These methods can clarify how different 14-3-3 isoforms achieve target specificity and how their heterodimerization fine-tunes stress responses.

Another pressing research objective is to investigate the crosstalk between 14-3-3-mediated salt stress signaling and other hormonal pathways, particularly ABA. Given that 14-3-3 proteins interact with ABA-responsive transcription factors and signaling components, a better understanding of their integration into broader hormonal networks will provide a more comprehensive view of plant adaptation to complex environmental stresses.

From a biotechnological perspective, the central role of 14-3-3 proteins in stress adaptation—with minimal impact on growth under normal conditions—makes them promising targets for improving crop salt tolerance. However, functional redundancy among isoforms may pose a challenge to effective manipulation. Overcoming this could involve simultaneous editing of multiple isoforms or targeting key 14-3-3–client interactions that govern critical salt tolerance pathways. Leveraging natural variation in 14-3-3 gene copy number and isoform function across crop wild relatives could also inform breeding strategies. Furthermore, leveraging natural or engineered splicing variants that enhance stress tolerance without compromising growth could offer a novel strategy for crop improvement. Harnessing 14-3-3 proteins and their interacting partners through genetic modifications or molecular breeding programs holds great potential for improving crop resilience to salinity, a major and increasing threat to global food security in the context of climate change.

## Figures and Tables

**Figure 1 plants-14-03568-f001:**
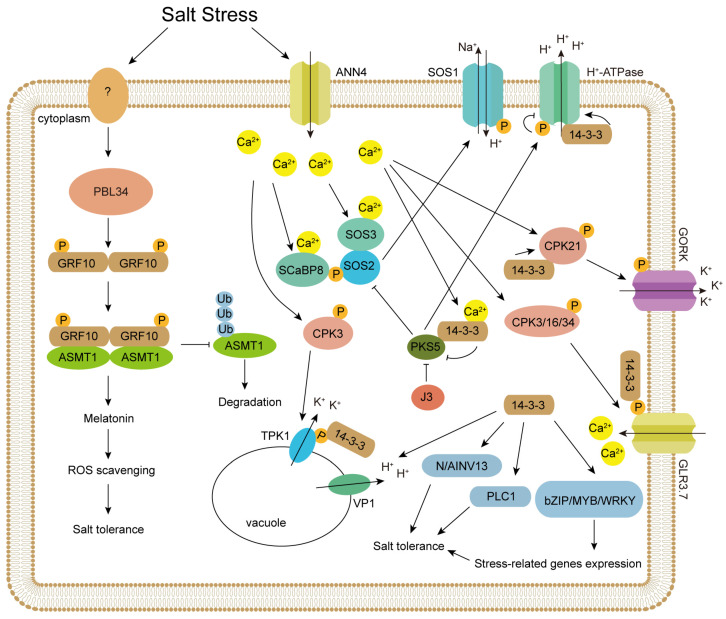
A schematic diagram of 14-3-3 protein regulating salt stress response in plants. Salt stress leads to an increase in intracellular Ca^2+^ concentration. Both SOS3 and SCaBP8 can bind to Ca^2+^, activating the kinase activity of SOS2. SOS2 phosphorylates SOS1, thereby activating its ion channel activity. The binding of 14-3-3 proteins to Ca^2+^ inhibits the kinase activity of PKS5, thereby relieving its inhibitory effects on SOS2 and H^+^-ATPase. The interaction between 14-3-3 proteins and H^+^-ATPase enhances its proton pump activity, while the phosphorylation of H^+^-ATPase by PKS5 inhibits the interaction between 14-3-3 proteins and H^+^-ATPase, consequently suppressing H^+^-ATPase activity. The increase in intracellular calcium ion concentration activates CDPKs. CPK3 phosphorylates TPK1, promoting its interaction with 14-3-3 proteins, thereby enhancing its channel activity and facilitating the pumping of potassium ions from the vacuole into the cytoplasm. 14-3-3 proteins interact with CPK21, enhancing its kinase activity. CPK21 phosphorylates GORK, increasing its channel activity and promoting potassium ion efflux. By facilitating the interaction between CPK21 and GORK, 14-3-3 proteins stabilize the ion channel activity of GORK. CPK3/16/34 promote the interaction between GLR3.7 and 14-3-3 proteins through phosphorylation, thereby enhancing its channel activity and facilitating the influx of calcium ions into the cell. 14-3-3 proteins play important roles in various plant species. Studies have shown that in *Nitraria sibirica*, 14-3-3 proteins interact with NsVP1 to enhance its proton pump activity, facilitating the transport of protons from the vacuole into the cytoplasm. 14-3-3 proteins enhance the enzymatic activities of N/AINV13 and PLC1, respectively, promoting related metabolic processes and improving plant salt stress tolerance. In plants, 14-3-3 proteins interact with transcription factors such as bZIP, MYB, and WRKY to enhance their transcriptional activity, thereby influencing the expression of stress-related genes and improving plant stress tolerance. Salt stress activates PBL34 via an unidentified receptor-like kinase. Subsequently, the activated PBL34 phosphorylates the 14-3-3 protein GRF10, which enhances its interaction with the melatonin synthase ASMT1 and inhibits the degradation of ASMT1. This promotes melatonin biosynthesis, clears excess reactive oxygen species, and improves salt tolerance in the plant.

## Data Availability

All data are included in this article.

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
