# Peer review of "Functional 14-3-3 Proteins: Master Regulators in Plant Responses to Salt Stress"

_plants, 2025, doi:10.3390/plants14233568_

Round 1
Reviewer 1 Report
Comments and Suggestions for Authors
General Assessment:
This manuscript presents a timely and comprehensive review on the crucial roles of 14-3-3 proteins in plant salt stress signaling. The authors effectively argue for the functions of 14-3-3 proteins as central signaling hubs and “molecular switches”. This is a very nice piece of work in synthesizing a vast amount of recent literature, systematically organizing the multifaceted interactions of 14-3-3 proteins with kinases, phosphatases, ion channels, proton pumps, enzymes, and transcription factors. The inclusion of a section on the regulation of 14-3-3 proteins themselves is a particular strength, adding depth to the discussion. Overall, this is a well-structured, informative, and valuable contribution to the field. However, I have a major concern and then some minor points.
Major Comments:
- Clarification of the "Release-of-Inhibition" Model: The "release-of-inhibition" model is a central theme (e.g., for SOS2, PKS5), but the mechanisms described sometimes seem contradictory. For instance, it is stated that salt stress disrupts the 14-3-3-SOS2 interaction, leading to SOS2 activation. However, it is also noted that salt stress promotes the 14-3-3-PKS5 interaction to repress PKS5. The authors should provide a clearer, more consolidated explanation in the discussion section on how these contrasting modes of action (disruption vs. promotion of interaction) both contribute to a coordinated stress response.
- Strengthening the "Future Perspectives": The "Conclusions and Future Perspectives" section is good but could be more forward-looking and specific.
The authors could elaborate on the techniques and approaches needed to address the knowledge gaps they identify. For example, how can isoform-specific functions be best studied (e.g., CRISPR-Cas9 mutants, isoform-specific interactomics )?
They could also discuss the potential for biotechnological application. Given that 14-3-3 proteins are master regulators, what are the challenges and opportunities in targeting them or their key interactions for improving crop salt tolerance? Is functional redundancy a major hurdle?
- The structure of Section 3 must be reorganized. The current layout lacks a clear logical progression and results in poor coherence. Furthermore, the title of Section 3 should be rephrased as a declarative statement, not a question. The phrasing of the titles "14-3-3 proteins are Regulated by..." is grammatically problematic as it creates a misleading passives voice construction. A clearer alternative would be "Regulation of 14-3-3 Proteins by..."
- The classification of "activated/inhibited" in Table 1 appears somewhat oversimplified. It would be beneficial to include more detailed descriptions of the underlying regulatory mechanisms.
Minor Comments:
- Lines 37-40: The final sentence "It also identifies critical knowledge gaps and outlines future research directions aimed at leveraging these proteins for improving crop resilience to salinity stress, an ongoing challenge in modern agriculture. conclusions." appears to have a typo ("conclusions."). Please revise for fluency.
- .Language and Flow: The manuscript is generally well-written. However, some sentences are very long and complex (e.g., the sentence spanning lines 165-172 on page 5). A thorough proofreading to break down overly long sentences would improve readability.
- Cross-referencing: In section 2.5, the text mentions enzymes like ASMT1, stating it is "further discussed in section 3.1." However, in section 3.1, the discussion of ASMT1 is embedded within the MdPBL34/MdGRF10 example. For clarity, it might be better to explicitly label the part of section 3.1 that covers ASMT1 or ensure the cross-reference is perfectly clear.
- When describe the impact of salt stress on crop productivity in Introduction (Line 70-98), the reference is too old. Please update the data and references, or delete the detailed description of numbers.
- Line 161, delete “s” in the word of “signals”.
- Line 131-134, the word "calcium" is repeated in the same sentence.
- Line 211, the word “interact” is incorrect.
- Line 213, "in a phosphorylation-dependent manner," "a" missing.
- Line 327, in the sentence “remains an open question that warrants further investigation”, delete “s” of word “question”.
Conclusion:
This is a thorough and insightful review that effectively consolidates current knowledge on the critical functions of 14-3-3 proteins in plant salt stress tolerance. The addition of a summary figure and the missing table, along with minor revisions to improve clarity and the future perspectives, will significantly enhance the manuscript's impact and usefulness to the readers of Plants. I recommend acceptance after these revisions.
Author Response
|
Comments 1: Clarification of the "Release-of-Inhibition" Model: The "release-of-inhibition" model is a central theme (e.g., for SOS2, PKS5), but the mechanisms described sometimes seem contradictory. For instance, it is stated that salt stress disrupts the 14-3-3-SOS2 interaction, leading to SOS2 activation. However, it is also noted that salt stress promotes the 14-3-3-PKS5 interaction to repress PKS5. The authors should provide a clearer, more consolidated explanation in the discussion section on how these contrasting modes of action (disruption vs. promotion of interaction) both contribute to a coordinated stress response. |
|
Response 1: Thank you for pointing this out. We agree with this comment. Therefore, in the discussion section, we delved deeper into this issue. Please refer to page 13, line 555-576 for details in the revised manuscript.
|
|
Comments 2: Strengthening the "Future Perspectives": The "Conclusions and Future Perspectives" section is good but could be more forward-looking and specific. The authors could elaborate on the techniques and approaches needed to address the knowledge gaps they identify. For example, how can isoform-specific functions be best studied (e.g., CRISPR-Cas9 mutants, isoform-specific interactomics )? They could also discuss the potential for biotechnological application. Given that 14-3-3 proteins are master regulators, what are the challenges and opportunities in targeting them or their key interactions for improving crop salt tolerance? Is functional redundancy a major hurdle? |
|
Response 2: Agree. We have revised the "Conclusions and Future Perspectives" sections according to your suggestions. Please refer to page 15, line 665-670 and 677-682 for details in the revised manuscript.
|
|
Comments 3: The structure of Section 3 must be reorganized. The current layout lacks a clear logical progression and results in poor coherence. Furthermore, the title of Section 3 should be rephrased as a declarative statement, not a question. The phrasing of the titles "14-3-3 proteins are Regulated by..." is grammatically problematic as it creates a misleading passives voice construction. A clearer alternative would be "Regulation of 14-3-3 Proteins by..." |
|
Response 3: Agree. We have revised the Section 3 according to your suggestions. Please refer to page 11-12 for details in the revised manuscript. |
|
|
|
Comments 4: The classification of "activated/inhibited" in Table 1 appears somewhat oversimplified. It would be beneficial to include more detailed descriptions of the underlying regulatory mechanisms. |
|
Response 4: We have supplemented Table 1 based on your suggestions. Please refer to page 23 for details in the revised manuscript. |
|
|
|
Comments 5: Lines 37-40: The final sentence "It also identifies critical knowledge gaps and outlines future research directions aimed at leveraging these proteins for improving crop resilience to salinity stress, an ongoing challenge in modern agriculture. conclusions." appears to have a typo ("conclusions."). Please revise for fluency. Response 5: The redundant word "conclusions" has been removed. Comments 6: Language and Flow: The manuscript is generally well-written. However, some sentences are very long and complex (e.g., the sentence spanning lines 165-172 on page 5). A thorough proofreading to break down overly long sentences would improve readability. Response 6: We conducted a thorough review of the manuscript and streamlined lengthy sentences to enhance readability. Comments 7: Cross-referencing: In section 2.5, the text mentions enzymes like ASMT1, stating it is "further discussed in section 3.1." However, in section 3.1, the discussion of ASMT1 is embedded within the MdPBL34/MdGRF10 example. For clarity, it might be better to explicitly label the part of section 3.1 that covers ASMT1 or ensure the cross-reference is perfectly clear. Response 7: We have changed the content in the parentheses to "ASMT1 is further discussed in section 3.1". Please refer to page 8, line 335-336 for details in the revised manuscript. Comments 8: When describe the impact of salt stress on crop productivity in Introduction (Line 70-98), the reference is too old. Please update the data and references, or delete the detailed description of numbers. Response 8: We have updated the data and references. Please refer to line 631-649 for details in the revised manuscript. Comments 9: Line 161, delete “s” in the word of “signals”. Response 9: We changed "calcium signals" to "calcium signal". Please refer to page 5, line 155 for details in the revised manuscript. Comments 10: Line 131-134, the word "calcium" is repeated in the same sentence. Response 10: We have rewritten the sentence to ensure that the word "calcium" is not repeated within the same sentence. Please refer to page 4, line 125-127 for details in the revised manuscript. Comments 11: Line 211, the word “interact” is incorrect. Response 11: We changed "interact" to "interacts". Please refer to page 5, line 200 for details in the revised manuscript. Comments 12: Line 213, "in a phosphorylation-dependent manner," "a" missing. Response 12: We added the missing "a". Please refer to page 5, line 201 for details in the revised manuscript. Comments 13: Line 327, in the sentence “remains an open question that warrants further investigation”, delete “s” of word “question”. Response 13: We deleted the “s” of word “question”. Please refer to page 8, line 311 for details in the revised manuscript. |

Reviewer 2 Report
Comments and Suggestions for Authors
The manuscript entitled “Function 14-3-3 Proteins: Master Regulators in Plant Responses to Salt Stress” is well-structured and contributes to our understanding of vital roles of 14-3-3 proteins in plant salt stress response. However, I would recommend addressing several points to strengthen the manuscript:
Major Comments:
1.While the abstract comprehensively covers the content of the review, it is currently too long and detailed for an abstract. The primary goal of an abstract is to provide a concise summary of the key findings and the article's significance. The current practice of listing numerous 14-3-3 interactors (e.g., SOS2, PKS5, CPK21, etc.) makes the text appear cluttered and distracts from the main message. I suggest the authors revise the abstract to be more concise, summarize the categories of interactors (e.g., kinases, transporters, transcription factors) without listing specific examples, and correct the grammatical errors. This will greatly improve readability and make a stronger first impression.
2.In Section 2, the authors summarize the primary types of proteins that interact with the 14-3-3 protein and detail the molecular mechanisms through which 14-3-3 proteins carry out their regulatory functions in salt stress tolerance. Nonetheless, the subtitles in this section need to be reorganized or rewritten, as their current wording is ambiguous.
3.In the two sections "3.1 14-3-3 proteins are regulated by protein kinase," and "3.2 14-3-3 proteins are regulated by ubiquitin ligase." Both instances pertain to post-translational modifications that regulate 14-3-3 proteins under salt stress. I believe there are additional mechanisms that regulate 14-3-3 proteins in response to salt stress, such as post-transcriptional modifications. This point requires further discussion.
4.In line 491, the phrase should be rephrased for clarity as "How are 14-3-3 proteins regulated in response to salt stress?"
Minor comments:
The text would benefit from careful proofreading and further polishing to address several grammatical issues. For example:
1.Line 499, revise "14-3-3 proteins are regulated by protein kinases" to "the regulation of 14-3-3 proteins by phosphorylation" for clarity.
2.Line 543, revise "14-3-3s are Regulated by Ubiquitin Ligases" to "the regulation of 14-3-3s by ubiquitination" for improved clarity.
3.Line 504, "14-3-3 proteins targets" should be "the target proteins of 14-3-3s" or "14-3-3 target proteins".
4.Line 507, "indicate" should be "indicates".
5.Line 527: The collocation "significantly high salt sensitivity" is grammatically problematic. A more appropriate phrasing would be "significantly increased salt sensitivity" or "heightened salt sensitivity".
6.Line 551, “remain” should be “remains”.
7.Line 548, “AtGCN4 overexpression plants” should be “AtGCN4-overexpressing plants”.
Author Response
|
Comments 1: While the abstract comprehensively covers the content of the review, it is currently too long and detailed for an abstract. The primary goal of an abstract is to provide a concise summary of the key findings and the article's significance. The current practice of listing numerous 14-3-3 interactors (e.g., SOS2, PKS5, CPK21, etc.) makes the text appear cluttered and distracts from the main message. I suggest the authors revise the abstract to be more concise, summarize the categories of interactors (e.g., kinases, transporters, transcription factors) without listing specific examples, and correct the grammatical errors. This will greatly improve readability and make a stronger first impression. |
|
Response 1: We revised and streamlined the abstract. Please refer to page 1 for details in the revised manuscript.
|
|
Comments 2: In Section 2, the authors summarize the primary types of proteins that interact with the 14-3-3 protein and detail the molecular mechanisms through which 14-3-3 proteins carry out their regulatory functions in salt stress tolerance. Nonetheless, the subtitles in this section need to be reorganized or rewritten, as their current wording is ambiguous. |
|
Response 2: We have reorganized the Section 2 according to your suggestions. The outline is as follows. Please refer to page 4-11 for details in the revised manuscript. 2.1 Reprogramming Stress Responses by Regulating Signaling Components 2.1.1 Regulation of Protein Kinases 2.1.2 Regulation of Protein Phosphatases 2.2 Maintaining Cellular Balance by Regulating Ion Homeostasis and Membrane Transport 2.2.1 Regulation of Ion Channels 2.2.2 Regulation of Proton Pumps 2.3 Coordinating Metabolism and Gene Expression by Regulating Metabolic Enzymes and Transcription Factors 2.3.1 Regulation of Key Metabolic Enzymes 2.3.2 Regulation of Transcription Factors |
|
Comments 3: In the two sections "3.1 14-3-3 proteins are regulated by protein kinase," and "3.2 14-3-3 proteins are regulated by ubiquitin ligase." Both instances pertain to post-translational modifications that regulate 14-3-3 proteins under salt stress. I believe there are additional mechanisms that regulate 14-3-3 proteins in response to salt stress, such as post-transcriptional modifications. This point requires further discussion. |
|
Response 3: We have addressed your suggestion by incorporating the information on 14-3-3 post-transcriptional modifications into the Discussion, Conclusion and Future perspectives sections. Please refer to page 12-15, line 592-609, 649-656, and 664-665 for details in the revised manuscript. |
|
|
|
Comments 4: In line 491, the phrase should be rephrased for clarity as "How are 14-3-3 proteins regulated in response to salt stress?" |
|
Response 4: We have corrected this sentence to “Regulation of 14-3-3 Proteins in Response to Salt Stress”. |
|
|
|
Comments 5: Line 499, revise "14-3-3 proteins are regulated by protein kinases" to "the regulation of 14-3-3 proteins by phosphorylation" for clarity. Response 5: We have corrected this sentence to “Regulation of 14-3-3 Proteins by Phosphorylation”. Comments 6: Line 543, revise "14-3-3s are Regulated by Ubiquitin Ligases" to "the regulation of 14-3-3s by ubiquitination" for improved clarity. Response 6: We have corrected this sentence to “The Regulation of 14-3-3s by Ubiquitination”. Comments 7: Line 504, "14-3-3 proteins targets" should be "the target proteins of 14-3-3s" or "14-3-3 target proteins". Response 7: We are grateful for your observation regarding this issue. We have addressed it and corrected all similar errors throughout the manuscript. Comments 8: Line 507, "indicate" should be "indicates". Response 8: We are grateful for your observation regarding this issue. We have addressed it and corrected all similar errors throughout the manuscript. Comments 9: Line 527: The collocation "significantly high salt sensitivity" is grammatically problematic. A more appropriate phrasing would be "significantly increased salt sensitivity" or "heightened salt sensitivity". Response 9: We are grateful for your observation regarding this issue. We have addressed it and corrected all similar errors throughout the manuscript. Comments 10: Line 551, “remain” should be “remains”. Response 10: We are grateful for your observation regarding this issue. We have addressed it and corrected all similar errors throughout the manuscript. Comments 11: Line 548, “AtGCN4 overexpression plants” should be “AtGCN4-overexpressing plants”. Response 11: We are grateful for your observation regarding this issue. We have addressed it and corrected all similar errors throughout the manuscript. |

Reviewer 3 Report
Comments and Suggestions for Authors
Dear Author
The article "Functional 14-3-3 Proteins: Master Regulators in Plant Re-2 Sponses to Salt Stress" was reviewed for scientific content and formatting. Based on existing literature, this review article summarizes the functions of 14-3-3 proteins under salt stress and their stress responses. Furthermore, any points that remain unclear are highlighted, guiding future research. Some typing errors were determined within the article. The article is not suitable for publication in its current form. However, it can be published after editing according to the recommended corrections.

Author Response
|
Comments 1: name of the species should be written Italic form. |
|
Response 1: We are grateful for your observation regarding this issue. We have addressed it and corrected all similar errors throughout the manuscript.
|
|
Comments 2: not only Na but also Cl is toxic for plant tissues. this information should be added. |
|
Response 2: We incorporated the relevant information about chloride ions in the introduction part based on your suggestion. Please refer to page 2, line 62-74 for details in the revised manuscript.
|
|
Comments 3: calcium (Ca2+) After the abbreviation of anything is given, it is better to use the abbreviation for homogeneity within the text! |
|
Response 3: We are grateful for your observation regarding this issue. We have addressed it and corrected all similar errors throughout the manuscript. |

Reviewer 4 Report
Comments and Suggestions for Authors
General comment
In this review, the authors provide a clear and well‐written report of how the 14-3-3 proteins interact with and regulate salt‐stress related proteins. However, the work would be much stronger if it included an evolutionary perspective across the plant kingdom. For example, recent studies demonstrate that the 14-3-3 family in angiosperms has undergone multiple rounds of duplication and loss, generating distinct sub-families and contributing to functional diversity in stress responses (see https://doi.org/10.3390/plants10122724; 10.1093/jxb/erad414; https://doi.org/10.1186/s12864-025-11513-0). Incorporating such evolutionary context would provide a broader view of 14-3-3 function, linking phylogenetic origin, gene‐family expansion/contraction, and diversification of regulatory interactions.
Minor comments
L38. There is a mistake at the end of the abstract; remove ‘conclusions.’
Latin names of the species must be in italics.
L85: remove ‘and binding’
L91: This statement, ‘The activation of SOS pathway is specific to NaCl stress and not triggered by other osmotic stressors such as KCl or mannitol’ is true for A. thaliana but has not been studied in all plant species, so it should be put into context.
Author Response

(The authors gave the same response as above.)

Round 2
Reviewer 4 Report
Comments and Suggestions for Authors
No further comments